# Neuroprotective Effect of Mixed Mushroom Mycelia Extract on Neurotoxicity and Neuroinflammation via Regulation of ROS-Induced Oxidative Stress in PC12 and BV2 Cells

**DOI:** 10.3390/cells14130977

**Published:** 2025-06-25

**Authors:** Sang-Seop Lee, Da-Hyun Ko, Ga-Young Lee, So-Yeon Kim, Seung-Yun Han, Jong-Yea Park, MiNa Park, Hyun-Min Kim, Ya-El Kim, Yung-Choon Yoo

**Affiliations:** 1Department of Microbiology, College of Medicine, Konyang University, Daejeon 32992, Republic of Korea; wgd.aria@gmail.com (S.-S.L.); goda1046@gmail.com (D.-H.K.); jc0012003@naver.com (G.-Y.L.); sksthdus09@hanmail.net (S.-Y.K.); 2Department of Anatomy, College of Medicine, Konyang University, Daejeon 32992, Republic of Korea; jjzzy@konyang.ac.kr; 3Giunchan Co., Ltd., Cheonan 31035, Republic of Korea; agnes223@daum.net (J.-Y.P.); guc2103@guc.co.kr (M.P.); guc2104@guc.co.kr (H.-M.K.); guc2105@guc.co.kr (Y.-E.K.)

**Keywords:** mushroom mycelial extract, excitotoxicity, neuroinflammation, PC12 neurons, BV2 microglia, ROS, NOX, NQO1, NRF2/HO-1 pathway, redox regulation, glutamate toxicity, neuroprotection, anti-inflammatory activity, apoptosis

## Abstract

In this study, we investigated the potential of a three-mushroom complex extract (GMK) to inhibit neuronal cell death induced by the activation of AMPA and NMDA receptors following glutamate treatment in NGF-differentiated PC12 neuronal cells. GMK significantly mitigated glutamate-induced excitotoxic neuronal apoptosis by reducing the elevated expression of BAX, a critical regulator of apoptosis, and restoring BCL2 levels. These neuroprotective effects were associated with redox regulation, as evidenced by the upregulation of SOD, CAT, and GSH levels, and the downregulation of MDA levels. Mechanistic studies further revealed that GMK effectively scavenged ROS by downregulating NOX1, NOX2, and NOX4, while upregulating NRF1, P62, NRF2, HO1, and NQO1. Additionally, in the same model, GMK treatment increased acetylcholine, choline acetyltransferase, and GABA levels while reducing acetylcholinesterase activity. These effects were also attributed to the regulation of redox balance. Furthermore, we investigated the antioxidant and anti-inflammatory mechanisms of GMK in LPS-stimulated BV2 microglia. GMK inhibited the activation of IκB and MAPK pathways, positively regulated the BCL2/BAX ratio, suppressed TXNIP activity, and upregulated NQO1 and NOX1. In conclusion, GMK improved neuronal excitotoxicity and microglial inflammation through the positive modulation of the redox regulatory system, demonstrating its potential as a natural resource for pharmaceutical applications and functional health foods.

## 1. Introduction

Aging and neurodegenerative diseases pose significant challenges in modern medicine [1]. A primary contributing factor to these diseases is the excessive generation of reactive oxygen species (ROS). ROS, including superoxide anion radical, hydroxyl radical, singlet oxygen, and hydrogen peroxide (H_2_O_2_), possess strong oxidative properties that induce oxidative stress in the human body [2]. The body’s primary antioxidants, such as superoxide dismutase (SOD), catalase (CAT), and glutathione peroxidase (GPx), serve as a defense mechanism to maintain homeostasis and protect against oxidative damage [3]. However, under pathological conditions, these protective systems may become insufficient, leaving the body vulnerable to oxidative stress [4]. This oxidative stress can lead to lipid peroxidation and cellular damage, ultimately contributing to the onset of various diseases, including cancer [5]. Alzheimer’s disease (AD), a representative neurodegenerative disease, is characterized by severe memory loss and cognitive decline [6]. Although its precise etiology remains unclear, a widely accepted hypothesis suggests that the extracellular deposition of amyloid-beta (Aβ) proteins and hyperphosphorylation of tau proteins disrupt synaptic connections between neurons, ultimately causing neuronal death [7,8]. This neuronal death is primarily driven by oxidative damage and inflammation induced by factors such as tau proteins, apolipoprotein E (ApoE), and amyloid precursor protein (APP) [9]. Oxidative stress in neurons is closely linked to redox reactions within living systems and is regulated by various factors, including ROS generation via NADPH oxidases (NOX), antioxidant activity mediated by the nuclear factor erythroid 2-related factor 2 (Nrf2)/heme oxygenase-1 (HO-1) signaling pathway, and regulation of NAD (P)H quinone oxidoreductase 1 (NQO1) [10,11,12]. Microglia, the immune cells of the central nervous system, play a critical role in neuroinflammation [13]. The activation of microglia into the M1 phenotype, triggered by pro-inflammatory factors such as tumor necrosis factor-α (TNF-α), interleukin-1 beta (IL-1β), and glutamate, promotes neurodegeneration [14]. Conversely, transitioning to the M2 phenotype, induced by anti-inflammatory factors such as interleukin-4 (IL-4) and gamma-aminobutyric acid (GABA), inhibits the formation of amyloid-beta plaques or directly removes them, thereby exhibiting neuroprotective effects [15,16]. Hence, maintaining sufficient antioxidant activity is crucial to prevent oxidative damage and the subsequent inflammation [17,18,19]. Mushrooms, consumed worldwide, are recognized as superfoods due to their diverse bioactive compounds that promote health [20]. Beta-glucan, a bioactive component found in mushrooms, enhances immune function by increasing natural killer (NK) cell activity, making it effective in cancer prevention [21]. Additionally, hericenone C-H and erinacine A-I, compounds found in the lion’s mane mushroom (Hericium erinaceus), promote the synthesis of nerve growth factor (NGF), thereby contributing to neuronal protection and improved learning abilities [22,23]. This study investigated the neuroprotective and anti-inflammatory mechanisms of a three-mushroom mycelial extract mixture (composed of *Phellinus linteus, Ganoderma lucidum*, and *Inonotus obliquus*) in glutamate-stimulated PC12 neuronal cells and LPS-stimulated BV2 microglial cells. Our previous studies on extracts derived from the mycelia of the same mushroom species as GMK demonstrated inhibitory effects on Aβ-induced neurotoxicity and focal cerebral ischemia (FCI). These findings suggested a potential involvement of antioxidant activity; however, the detailed mechanisms remained unexplored [24,25]. In this study, we aimed to evaluate the potential of this mixture through mechanistic research focusing on redox regulation.

## 2. Materials and Methods

### 2.1. GMK Preparation

GMK was prepared as follows. Mixed mycelium of *Phellinus linteus*, *Inonotus obliquus*, and *Ganoderma lucidum* cultured in liquid medium was inoculated into sterilized barley solid medium at 3% (*w*/*w*). This mushroom mycelium complex was co-cultured for 50–55 days under conditions of 40–60% humidity and 25–30 °C, then dried at 57–60 °C for 24 h. The culture and purified water were mixed in a 1:10 ratio and extracted at 75–80 °C. After removing the precipitate through filtration, the extract was concentrated at 55–60 °C and then freeze-dried (GMK). GMK was stored at −20 °C until use.

### 2.2. LC-MS/MS Analysis and Metabolite Identification

The compositional analysis of GMK was performed using an unbiased metabolomics analysis ultra-performance liquid chromatography (UPLC) system (Waters, Milford, CT, USA). The chromatographic separation was carried out using an ACQUITY UPLC HSS T3 column (100 mm × 2.1 mm, 1.8 μm, Waters) with a column temperature of 40 °C and a flow rate of 0.5 mL/min, where the mobile phase contained solvent A (water +0.1% formic acid) and solvent B (acetonitrile +0.1% formic acid). Metabolites were eluted using the following gradient elution conditions: 97% phase A for 0–5 min; 3–100% liner gradient phase B for 5~16 min; 100% phase B for 16–17 min; 100–3% reverse liner gradient phase B for 17~19 min; 97% Phase A for 19–25 min. The loading volume of each sample was 5 μL. The metabolites eluted from the column were detected by a high-resolution tandem mass spectrometer SYNAPT XS QTOF (Waters) in positive and negative ion modes. For positive ion mode, the capillary voltage and the cone voltage were set at 2 kV and 40 V, respectively. For negative ion mode, they were 1 kV and 40 V, respectively. Centroid MS^E^ mode was used to collect the mass spectrometry data. The primary scan ranged from 50 to 1200 Da and the scanning time was 0.2 s. All the parent ions were fragmented using 20–40 eV. The information of all fragments were collected and the time was 0.2 s. In the data acquisition process, the LE signal was gained every 3 s for real-time quality correction. For accurate mass acquisition, leucine enkephalin at a flow rate of 10 μL min^−1^ was used as a lock mass by a lock spray interface to monitor the positive ([M + H]^+^ = 556.2771) and the negative ([M − H]^−^ = 554.2615) ion modes. Raw mass spectrometry data were processed and extracted, and peaks identified using commercial software progenesis QI (version 2.2; Waters Corporation, Milford, MA, USA) implementation. The metabolites were filtered on the basis of the ANOVA *p* value < 0.05 and max fold change ≥ 2. Potential markers of interest were extracted from an S-plot (EZinfo software 2.0) constructed following partial least squares-discriminant analysis (PLS-DA), and markers were chosen based on their contribution to the variation and correlation within the data set. With regard to the identification of biomarkers, the ion spectrum of potential biomarkers was matched with the structure message of metabolites acquired from available biochemical databases, such as HMDB (http://www.hmdb.ca/, accessed 17 April 2025), METLIN (http://metlin.scripps.edu/, accessed 17 April 2025), MassBank (http://www.massbank.jp/, accessed 17 April 2025), and Lipidmaps (http://www.lipidmaps.org/, accessed 17 April 2025). The reconstruction, interaction, and pathway analysis of potential biomarkers was performed with MetPA (MetaboAnalyst; http://www.metaboanalyst.ca/, accessed 1 May 2025) based above database source to identify the metabolic pathways. MetPA uses the high-quality KEGG metabolic pathways as the backend knowledgebase.

### 2.3. In Silico Annotation and Prioritization of Metabolites

Initially, all tentatively identified compounds (n > 30) were subjected to manual literature investigation. Compounds previously reported in mushrooms and barley or likely to occur in these sources were selected based on the available scientific literature. Additionally, only compounds demonstrated to have antioxidant, anti-inflammatory, or neuroprotective activities were subjected to in silico annotation and prioritization process. The structural identity of the metabolites selected through spectral library matching was verified by GNPS (Global Natural Products Social Molecular Networking). Matches exhibiting a threshold of ≥0.7 on the cosine similarity score criterion were considered reliable. For spectra without a reliable GNPS hit, MS-fragmentation scoring was performed in both MetFrag and SIRIUS (CSI:FingerID) against KEGG, PubChem, MetaCyc, HMDB and ChEB. Candidates reaching a MetFrag score ≥ 70% or a CSI:FingerID scoring ≥ 70% were retained as biologically relevant.

### 2.4. GMK Treatment Concentration Design

All GMK concentrations used in this study were selected to maintain ≥ 80% cell viability in vitro testing. This viability benchmark was consistently applied during MTT-based prescreening and subsequently adapted to suit the analytical requirements of each assay (e.g., ELISA dynamic range, mechanistic target validation, or inhibitory window coverage).

In preliminary MTT assays (Figure 1A), GMK exhibited no cytotoxicity up to 200 µg/mL in both undifferentiated and NGF-differentiated PC12 cells. Notably, only the 100 and 200 µg/mL concentrations significantly attenuated glutamate-induced excitotoxicity in differentiated cells. Based on these findings, subsequent mechanistic experiments in PC12 cells—such as TUNEL staining, intracellular ROS quantification (FACS), and Western blotting—were conducted using 100 and 200 µg/mL as effective, non-cytotoxic doses.

In contrast, cell-based antioxidant and neurotransmitter assays (SOD, CAT, GSH, MDA, acetylcholine, GABA, etc.) were performed using a broader concentration range (25–200 µg/mL), reflecting the requirements of ELISA-based quantification, which often demands a wider dynamic range to detect subtle dose-dependent effects.

BV2-based assays employed an even broader range (31.25–500 µg/mL), as no significant cytotoxicity was observed up to 500 µg/mL, and all doses met the ≥80% viability criterion defined by ISO 10993-5. This allowed for the full coverage of the inhibitory window in ELISA-based analyses of NO, TNF-α, and IL-6. Western blotting was performed at 100 and 200 µg/mL across both PC12 and BV2 models to facilitate cross-comparison of mechanistic outcomes.

In cell-free antioxidant assays (DPPH, ABTS, and reducing power), GMK was tested at a broader concentration range (up to 500 µg/mL), as these assays are not limited by cytotoxicity concerns. The applied concentrations were selected to encompass the full dynamic range of radical scavenging and reducing capacity, and to enable direct comparison with the positive control (BHT).

### 2.5. Positive Control Reagents

N-acetyl-cysteine (A9165, Sigma-Aldrich, St. Louis, MO, USA), memantine (PHR1886, Merck, Rahway, NJ, USA), and donepezil (4385, R&D Systems, Minneapolis, MN, USA) were used according to each experimental protocol. Other reagents were purchased from Sigma-Aldrich. N-acetyl-cysteine, a precursor of glutathione, has been reported to have symptom-suppressive and preventive effects against various neurological disorders related to neuronal excitotoxicity through glutathione-mediated redox regulation [26,27]. In the glutamate-stimulated PC12 neuron model, the non-competitive NMDA receptor antagonist memantine and the acetylcholine esterase inhibitor donepezil were used as positive controls. Memantine, used for moderate to severe dementia, is known to significantly inhibit glutamate-induced excitotoxicity and neuronal cell death [28]. Donepezil, primarily used for mild to moderate dementia, does not directly interact with NMDA or AMPA receptors but has been reported to suppress excitatory neuronal cytotoxicity through mechanisms such as reducing Ca^2^⁺ influx into neurons caused by glutamate [29].

### 2.6. MTT Assay

PC12 cells (ATCC, Manassas, VA, USA) were plated in a 48-well plate with DMEM medium (LM001-01, WELGENE, Gyeongsan, Republic of Korea) containing 10% FBS (16000-044, Thermo, Waltham, MA, USA). Cells were stabilized in a 5% CO_2_ incubator for 6 h. Following stabilization, 5 mM glutamate (Sigma-Aldrich) and GMK were co-treated for 24 h, after which the MTT assay was performed. Absorbance was measured at 540 nm using a BIO-RAD iMark Microplate Reader (Bio-Rad Laboratories, Hercules, CA, USA).

### 2.7. TUNEL and Hoechst Staining

PC12 cells were co-treated with 5 mM glutamate and GMK for 24 h. Cells were then washed twice with PBS and fixed with 4% PFA for 30 min. Following fixation, cells were washed with PBS and stained with Hoechst 33258 (2 μg/mL) for 30 min in the dark at 37 °C. For TUNEL staining, the DeadEnd™ Fluorometric TUNEL System (Promega, Fitchburg, WI, USA) was applied following the same protocol. Stained cells were observed using an LSM 900 Confocal Microscope (ZEISS, Oberkochen, Germany).

### 2.8. ROS Measurement

To measure ROS levels, cells were treated with DCF-DA (chloromethyl-2′,7′-dichlorodihydrofluorescein diacetate, CM-H2DCFDA). PC12 cells were washed with cold PBS, followed by treatment with 15 μM DCF-DA (Thermo) in the dark for 30 min. ROS levels were measured at 538 nm using a BD FACSCalibur (BD Biosciences, Franklin Lakes, NJ, USA).

### 2.9. Measurement of Antioxidant Enzymes and MDA

PC12 cells were co-treated with 5 mM glutamate and GMK for 24 h. The supernatant was removed, and cells were washed twice with PBS, followed by lysis with RIPA buffer (Thermo). The lysate was quantified, and levels of antioxidant enzymes were measured using EZ-SOD, EZ-Catalase, and EZ-Glutathione assay kits (DoGenBio, Seoul, Republic of Korea). Lipid peroxidation was assessed by measuring MDA levels using a Lipid Peroxidation (MDA) Assay Kit (Sigma-Aldrich).

### 2.10. Western Blot Analysis

PC12 cells cultured under the same conditions as in Section 2.6 were lysed, and 20 μg of protein was loaded for SDS-PAGE electrophoresis. After transfer to a PVDF membrane, the blots were probed with primary antibodies. Secondary antibodies conjugated with HRP were applied, and detection was performed using a Chemiluminescence Imaging System (Azure Biosystems, Dublin, CA, USA). For BV2 cells, 1 × 10^6^ cells/well were seeded in a 6-well plate and stabilized for 6 h, followed by co-treatment with GMK at various concentrations and LPS (100 ng/mL) for 120 min [30]. Proteins (20 μg per sample) were subjected to SDS-PAGE, and blots were probed with target antibodies, followed by HRP-conjugated secondary antibodies. Signals were detected using a Chemiluminescence Imaging System (Azure Biosystems). Antibodies for NOX1, NOX2, NOX4, NQO1, BCL2, and BAX were obtained from Elabscience (Houston, TX, USA), while antibodies for NRF1, NRF2, HO1, P62, TXNIP, TRX1, phospho-ERK, ERK, phospho-JNK, JNK, phospho-P38, and P38 were purchased from Cell Signaling Technology (Danvers, MA, USA).

### 2.11. ELISA and NO Assay in BV2 Cells

BV2 cells (ATCC) were seeded at a density of 1 × 10^4^ cells/well in a 96-well plate, stabilized for 6 h, and treated with various concentrations of GMK overnight. After treatment, 100 ng/mL of LPS was added, and 24 h later, the supernatant was collected. NO levels were measured using the Griess Reagent (Sigma-Aldrich). TNF-alpha and IL-6 levels were quantified using the Mouse TNF-alpha DuoSet ELISA Kit (R&D Systems) and the BD OptEIA™ Mouse IL-6 ELISA Set (BD Biosciences), respectively.

### 2.12. Measurement of Neurotransmitters

Differentiated PC12 cells were co-treated with 5 mM glutamate and GMK for 24 h. Following incubation, neurotransmitter levels in the culture medium were assessed using the Rat Acetylcholine ELISA Kit, Rat Choline Acetyltransferase ELISA Kit, Rat Acetylcholinesterase ELISA Kit, and Rat Gamma-Aminobutyric Acid ELISA Kit (MyBioSource, San Diego, CA, USA).

### 2.13. DPPH Radical-Scavenging Assay

Initially, 50 μL of the sample extract was mixed with 450 μL of DPPH reagent (200 μM) dissolved in methanol. The mixture was incubated in the dark at room temperature for 20 min, and the absorbance was measured at 570 nm. The DPPH radical-scavenging activity was calculated using the following formula:DPPH-Scavenging Activity (%) = {(A − B)/C} × 100
where A = absorbance of sample + DPPH, B = absorbance of methanol + DPPH, and C = absorbance of sample + methanol. Butylated hydroxytoluene (BHT) was used as the positive control.

### 2.14. ABTS Radical-Scavenging Assay

A mixture of 7.4 mM ABTS and 2.6 mM potassium persulfate (K_2_S_2_O_8_) was prepared and incubated in the dark at room temperature for 16 h. The solution was diluted with methanol to achieve an absorbance of 1.0 ± 0.05 at 730 nm before use. For the assay, 20 μL of the sample was mixed with 980 μL of the ABTS reagent and allowed to react at 30 °C for 7 min. Absorbance was measured at 730 nm. The ABTS radical-scavenging activity was calculated as the percentage reduction in absorbance of the sample compared to the blank.

### 2.15. Reducing Power Assay

To 1 mL of the GMK sample at varying concentrations, 1 mL of 1% potassium ferricyanide was added, and the mixture was incubated at 50 °C for 20 min. Subsequently, 1 mL of 10% trichloroacetic acid was added to the reaction mixture and centrifuged at 7000× *g* for 10 min. The supernatant was mixed with an equal volume of distilled water and 1/10 volume of 0.1% ferric chloride solution. Absorbance was measured at 730 nm, and the results were expressed as concentration-dependent optical density (OD) values.

### 2.16. Statistical Analysis

All statistical analyses were performed using IBM SPSS Statistics 25 and SAS 9.4 software. Student’s *t*-test and ANOVA were conducted with confidence intervals of * *p* < 0.05, ** *p* < 0.01, and *** *p* < 0.001.

## 3. Results

### 3.1. Inhibitory Activity of GMK Against Glutamate-Induced Neurotoxicity

The excitatory neurotransmitter glutamate is known to mediate cell death through excitotoxicity by acting on NMDA, AMPA, and kainate receptors when excessively re-leased [31,32]. To investigate the inhibitory effects of GMK on glutamate-induced excitotoxicity in PC12 cells, we performed MTT assays (Figure 1). The results showed that GMK effectively suppressed glutamate-induced excitotoxicity in PC12 cells.

### 3.2. Apoptosis Inhibition Activity Through Regulation of the BCL-2/BAX Pathway

Using the same model, Hoechst 33342 and TUNEL staining were performed (Figure 2A). The findings indicated that GMK exhibited significant cell-protective activity, effectively inhibiting glutamate-induced apoptosis in a concentration-dependent manner. Analysis of apoptosis-related signaling pathways at the protein level revealed that GMK upregulated BCL-2 and P62 while downregulating BAX in a concentration-dependent manner (Figure 2B). Glutamate-induced excitotoxicity causes excessive Ca^2^⁺ influx and the disruption of mitochondrial homeostasis [33,34,35]. Upregulation of BAX, along with the downregulation of BCL-2, mediates the activation of cytochrome C and APAF1, which subsequently activates caspase-9 and caspase-3, leading to apoptosis [36]. These results suggest that GMK exerts neuroprotective effects by modulating the BCL-2/BAX axis.

### 3.3. ROS-Scavenging Activity

Glutamate-induced excitotoxicity is closely associated with oxidative damage caused by ROS, which is linked to cystine regulation and a decrease in glutathione (GSH) levels [37,38]. Based on this perspective, we analyzed the ROS-scavenging activity of GMK using FACS after treating glutamate-treated PC12 cells with DCF-DA (Figure 3). N-acetyl-cysteine was used as a control. The results indicated that ROS levels were reduced in a concentration-dependent manner by GMK treatment.

### 3.4. Reductive Activity of GMK in Western Blot

Underlying the neuronal damage associated with neuroexcitotoxicity is oxidative stress and oxidative damage caused by ROS and RNS [39,40]. From this perspective, an analysis of oxidation reduction reaction relation to signaling at the protein level revealed that GMK effectively reduces oxidative stress in neurons through its ROS-suppressing mechanism via the downregulation of NOX1, NOX2, and NOX4, along with redox activity mediated through the regulation of the NQO1, NRF1, NRF2, and HO1 in PC12 cells (Figure 4). They are known to be important regulators of redox balance [41,42,43,44,45].

### 3.5. Measurement of Intracellular SOD, CAT, and GSH Levels

Molecules involved in redox reactions mediate redox regulation through the activity of key antioxidants, including superoxide dismutase (SOD), catalase (CAT), glutathione (GSH), and peroxiredoxin (PRX) [46]. To evaluate GMK’s antioxidant effects, the levels of SOD, CAT, and GSH were measured using the same cellular model (Figure 5). The GMK hot water extract exhibited antioxidant effects, though not as strong as those of N-acetyl-cysteine (NAC) in terms of SOD and CAT activity. However, it demonstrated dose-dependent antioxidant activity comparable to that of memantine and donepezil. In the measurements of malondialdehyde (MDA) and GSH levels, GMK showed a significant increase, although its effectiveness was lower than that of the positive controls.

### 3.6. DPPH, ABTS, and Reducing Power

As a complementary investigation of in vitro antioxidant activity, the antioxidant potential of GMK was assessed using DPPH, ABTS, and reducing power assays. The results demonstrated that the GMK extract exhibited antioxidant activity comparable to that of the positive control, butylated hydroxytoluene (BHT), across all three methods (Figure 6).

### 3.7. Measurement of Three Acetylcholine-Related Indicators and GABA Levels

Significant decreases in acetylcholine activity are commonly observed in neurodegenerative diseases, including Alzheimer’s disease [47]. Oxidative stress is known to inhibit acetylcholine activity by regulating acetylcholinesterase (AchE) and choline acetyltransferase (ChAT) [48]. Although the inhibitory neurotransmitter gamma-aminobutyric acid (GABA) is not directly involved in neurodegenerative diseases, it can inhibit glutamate-induced excitotoxicity through the neuronal glutamate–GABA–glutamine metabolic system. GABA is synthesized from glutamate via decarboxylation, catalyzed by glutamic acid decarboxylase (GAD). Upregulation of GABA levels generally leads to a decrease in glutamate levels. GABA participates in the glutamate–GABA–glutamine cycle, which involves astrocytes, glutamatergic and GABAergic neurons, and the tricarboxylic acid (TCA) cycle [49,50].

To determine whether the redox regulatory mechanism of GMK that inhibits glutamate-induced neuronal excitotoxicity is reflected in neurotransmitter activity, we measured acetylcholine, acetylcholinesterase, choline acetyltransferase, and GABA levels (Figure 7). The NMDA receptor antagonist memantine and the acetylcholinesterase inhibitor donepezil were used as positive controls.

The results demonstrated that GMK extract, while less potent than the controls, significantly improved activity across all indicators. These findings suggest that GMK inhibits glutamate-induced excitatory neurotoxicity through its antioxidant activity, and that this mechanism is associated with increased acetylcholine and GABA activity.

### 3.8. Anti-Inflammatory Activity of GMK in Microglial Cells

Neurotoxicity and microglial inflammatory responses are closely interconnected, with microglia mediating early steps in the development of glutamate-induced neuroexcitotoxicity [51,52]. To investigate the anti-inflammatory effects of GMK, we used an LPS-stimulated BV2 cell model (Figure 8). The results demonstrated that GMK significantly reduced the levels of NO, TNF-α, and IL-6. Western blot analysis revealed that GMK efficiently suppressed the phosphorylation of JNK, ERK, p38, and IκB. Furthermore, GMK was found to upregulate BCL2 and downregulate BAX, thereby inhibiting apoptosis associated with the inflammatory response in microglia. These findings suggest that GMK efficiently suppresses the inflammatory response of microglia, likely through antioxidant mechanisms involving the downregulation of NOX and TXNIP, as well as the upregulation of NQO1.

### 3.9. Tentative Chemical Composition Analysis of GMK

The chemical composition of GMK was analyzed to expect and identify potential biologically active components responsible for the observed effects. Various metabolites, including glucosylceramides [53], triterpenoids [54], indole alkaloids [55], phenolic glycosides [56], and sterols [57], was identified by UHPLC-QTOF-MS/MS analysis. Compounds of these families have been reported from various mushroom species or mushroom cultivation media, and are considered to be potent bioactive components contributing to regulation of inflammatory, oxidation, neurodegenerative disease, and dementia. Therefore, all selected candidate compounds were screened based on previous reports of antioxidant, anti-inflammatory, and neuroprotective activities and on previous identification in mushrooms or their potential presence in mushroom-derived substances. Furthermore, only compounds with a similarity score ≥ 0.85 as determined by GNPS [58] matching or MetFrag [59], SIRIUS [60] in silico analysis were considered. As shown in Figure 9, among the identified compounds of GMK, N-(2-methylamino-benzoyl) tryptamine, detected in the ES⁺ mode, is a tryptamine derivative known for its potential neuromodulatory and neuroprotective activities [61,62]. In addition, pinellic acid, identified in the ES⁻ mode, has been previously reported to exhibit anti-inflammatory and antioxidant effects [63]. These findings further support the biological relevance of the compounds tentatively identified in GMK.

## 4. Discussion

### 4.1. Redox Imbalance and NOX-Mediated Excitotoxicity

According to physical principles, matter tends to remain in its most stable state [64]. Reactive oxygen species (ROS) are unstable oxygen molecules with an excited electronic configuration that seek to stabilize by stealing electrons from surrounding molecules. This process results in oxidative damage, commonly referred to as oxidative stress [65]. Consequently, antioxidants play a critical role as the first line of defense in protecting the body against oxidative stress induced by ROS. ROS are generated by NADPH oxidases (NOX), a family of enzymes that produce superoxide and other ROS by transferring electrons across the cell membrane [66,67]. In brain tissue, ROS generated by NOX1-4 play a crucial role in the degenerative changes associated with neuronal death [68]. Across the entire spectrum of cognitive decline, from mild cognitive impairment to severe Alzheimer’s disease, NOX-induced ROS production has been identified as a key mechanism contributing to oxidative stress [69].

In this study, GMK treatment downregulated the expression of NOX1, NOX2, and NOX4 at the protein level while upregulating the BCL-2/BAX ratio. These findings suggest that GMK effectively inhibits cell death in PC12 cells caused by glutamate-induced excitotoxicity [70].

### 4.2. Activation of Phase-II Antioxidant Defenses

The NAD⁺/NADH ratio, regulated by NAD(*P*)H: Quinone Oxidoreductase 1 (NQO1), is closely linked to aging and neurodegenerative diseases [71]. NQO1 downregulates NOX activity, thereby reducing oxidative stress [72]. The NRF2/HO-1 pathway is another critical regulator of redox homeostasis. In its inactive form, NRF2 is bound to Keap1 in the cytoplasm. Upon exposure to oxidative stress or ROS production, NRF2 dissociates from Keap1 and translocates to the nucleus. In the nucleus, NRF2 forms a heterodimer with small Maf proteins and binds to antioxidant response elements (AREs), activating the expression of antioxidant genes, including heme oxygenase-1 (HO-1), which initiates cellular defense mechanisms [10,73,74].

In this study, GMK exhibited antioxidant activity by upregulating phase II antioxidant enzymes such as NQO1 and HO-1. This effect was further supported by increased levels of endogenous antioxidants, including SOD, CAT, and GSH, and a concomitant decrease in lipid peroxidation, as indicated by reduced MDA levels.

### 4.3. NRF1-Linked Regulation and Neurotransmitter Homeostasis

NRF1, essential for maintaining mitochondrial homeostasis, interacts with NRF2, HO-1, and other factors to regulate redox reactions and energy metabolism [75]. Additionally, NRF1 induces P62 and GABARAPL1 (GABA Type A Receptor Associated Protein Like 1), which are involved in autophagy regulation [76]. GMK was shown to upregulate NRF1, P62, and GABA, suggesting its potential antioxidant and inhibitory neurotransmission activities mediated by GABA [77] (Figure 10A).

Based on these results, we plan to conduct further research to explore the potential of GMK in preventing and mitigating Alzheimer’s disease. This study will focus on the NRF1/P62/GABARAPL1 regulatory mechanism of GMK and its ability to enhance astrocyte metabolism through the glutamate/GABA/glutamine cycle [50,78,79].

### 4.4. Modulation of Glutamatergic Signaling

When glutamate released from the presynaptic neuron enters the synaptic cleft and binds to AMPA receptors, Na⁺ ions flow into the postsynaptic neuron, causing depolarization [80]. This depolarization displaces Mg^2^⁺ ions, which normally block NMDA receptor activity [81]. When glutamate subsequently binds to NMDA receptors, excitatory neurotransmission is triggered [82]. Activation of both AMPA and NMDA receptors increases ROS production and, conversely, can also be induced by ROS [83,84]. Acetylcholine, acting on both nicotinic and muscarinic receptors, is closely associated with learning and memory functions, and a decrease in acetylcholine is correlated with neuronal cell death caused by excitotoxicity [85].

In this study, SOD, CAT, and GSH levels were positively correlated with acetylcholine and choline acetyltransferase levels, while showing an inverse correlation with acetylcholinesterase and MDA levels. Comparatively, the NMDA antagonist memantine, the acetylcholinesterase inhibitor donepezil, and the glutathione precursor N-acetylcysteine exhibited activities similar to GMK [86,87]. These findings suggest that the glutamate- and acetylcholine-regulated mechanisms of GMK are closely related to redox-regulated responses [88].

### 4.5. Suppression of Microglial TLR4-MAPK Inflammation

Neuronal cell death and microglial inflammatory responses are closely interconnected [89]. The TLR4 inflammatory response in microglia can induce inflammatory activation of astrocytes and disrupt the glutamate–GABA–glutamine metabolic system [90,91]. GMK was observed to suppress the TLR4 inflammatory response in microglial cells by reducing the phosphorylation of MAPK and IκB, while also preventing cell death by regulating the BCL2/BAX ratio [92]. From the perspective of redox regulation, GMK appears to suppress the TLR4 inflammatory response through thioredoxin-mediated antioxidant activity, particularly by reducing TXNIP levels [93,94,95,96].

### 4.6. Synergy of the Three-Mushroom Blend

Each of the three mushroom extracts that make up GMK has been reported to exhibit activity in inhibiting nerve damage or degeneration. However, detailed studies based on broader redox mechanisms have not yet been performed [97,98,99]. This study is significant in that it focuses on the synergistic effects of the three mushroom extracts while simultaneously identifying a comprehensive redox regulation mechanism [100,101], centered on the downregulation of NOX and the upregulation of NQO1 [102] (Figure 10B). Furthermore, the rationale for combining *Inonotus obliquus*, *Ganoderma lucidum*, and *Phellinus linteus* in the GMK formulation is grounded in numerous previous studies reporting their individual antioxidant, anti-inflammatory, and neuroprotective activities [24,25,103,104,105]. Rather than assessing each species in isolation, the present study focused on their potential synergistic effects within a standardized extract.

Considering these results, GMK demonstrates a comprehensive mechanism of action that suppresses microglial inflammatory responses through redox regulation. It also efficiently reduces glutamate-induced excitatory neurotoxicity by upregulating acetylcholine and GABA, exhibiting significant neuroprotective activity [106]. In conclusion, these findings suggest that GMK possesses substantial potential as a mushroom-derived natural pharmaceutical material, with its effects centered on antioxidant and anti-inflammatory activities.

### 4.7. Limitations and Future Directions

Although this study provides compelling in vitro evidence for the neuroprotective and anti-inflammatory activity of GMK, the precise identification of its active components remains a critical next step. Using UHPLC-QTOF-MS/MS analysis coupled with in silico tools such as GNPS, MetFrag, and SIRIUS, several high-confidence Level 2 candidate compounds have been annotated. Future studies will focus on isolating these constituents to achieve Level 1 identification, enabling rigorous structure–activity relationship (SAR) analysis. Subsequently, the pharmacodynamic efficacy of these purified compounds will be validated in vivo using relevant animal models of neuroinflammation and oxidative brain injury, with a mechanistic focus on NOX isoforms, TXNIP/TRX/NF-κB pathway, MAPK/NF-κB pathway, and the NRF2/HO-1 axis. In addition, while current in vitro data suggest a favorable safety profile, further in vivo studies are needed to evaluate long-term safety. These efforts are expected to clarify the bioactive basis of GMK and support its development as a targeted redox-modulatory agent for neurodegenerative diseases.

## Figures and Tables

**Figure 1 cells-14-00977-f001:**
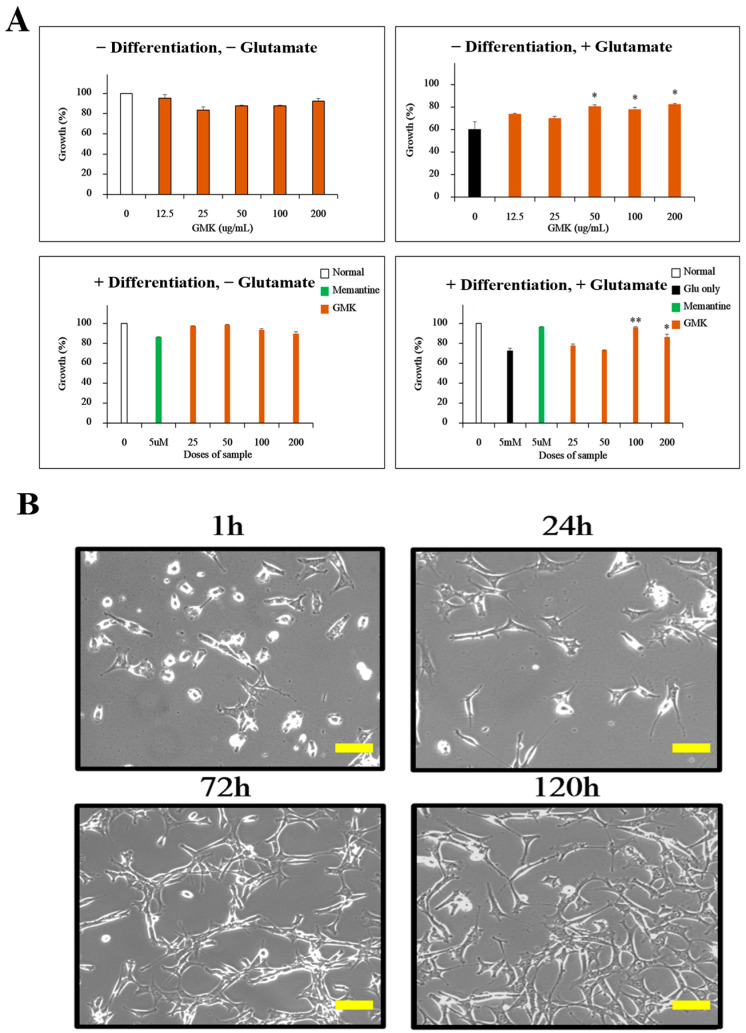
Inhibition of excitotoxicity by GMK in PC12 cells. (**A**) Inhibitory effects of GMK on glutamate-induced excitotoxicity in both differentiated and undifferentiated PC12 cells. (**B**) Microscopic images showing PC12 cell differentiation over time after NGF treatment. GMK exhibited concentration-dependent inhibition of excitotoxicity. Scale bar = 20 μm. Values are expressed as the mean ± SD. * *p* < 0.05 compared with the Glu-only group, ** *p* < 0.01 compared with the Glu-only group.

**Figure 2 cells-14-00977-f002:**
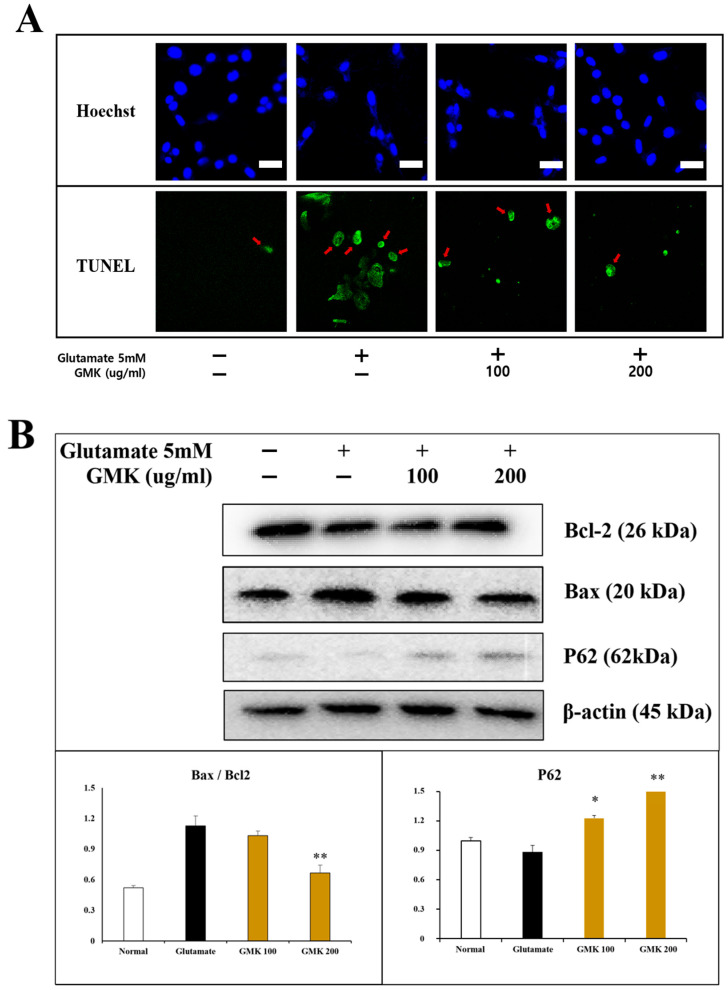
Apoptosis inhibitory activity in cell staining (**A**) and Western blot (**B**). Results from Hoechst 33342 and TUNEL staining demonstrated that GMK strongly inhibited cell apoptosis caused by glutamate-induced excitotoxicity. GMK exhibited antiapoptotic mechanisms by upregulating BCL2 and downregulating BAX. The increased P62 expression and BCL2/BAX ratio were statistically significant. Scale bar in (**A**) = 20 μm. Values are expressed as the mean ± SD from three independent measurements. * *p* < 0.05 compared with the glutamate-only group, ** *p* < 0.01 compared with the glutamate-only group.

**Figure 3 cells-14-00977-f003:**
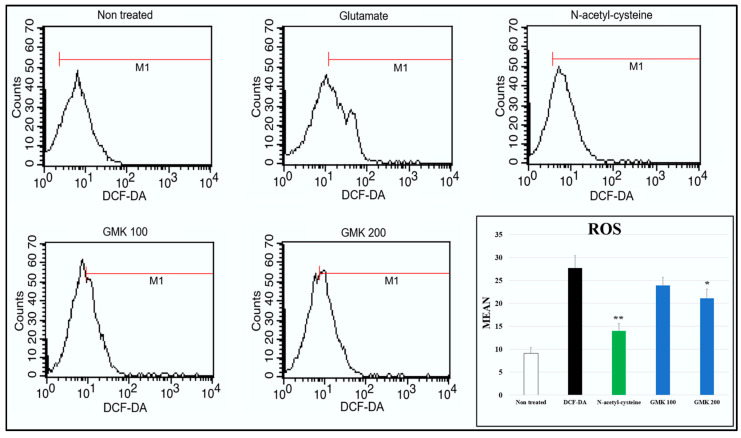
ROS-scavenging activity measured by FACs. ROS levels increased approximately 2.9-fold following glutamate treatment, and GMK treatment reduced ROS levels in a concentration-dependent manner. Values are expressed as the mean ± SD. * *p* < 0.05 compared with the glutamate group, ** *p* < 0.01 compared with glutamate group.

**Figure 4 cells-14-00977-f004:**
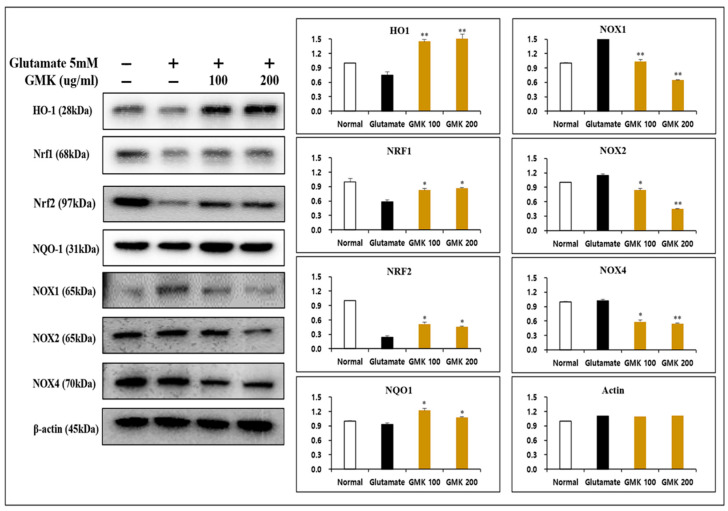
Reductive activity of GMK in Western blot. GMK significantly decreased the expression of NOX1, NOX2, and NOX4 while upregulating NQO1, NRF1, and HO1 in glutamate-treated PC12 cells. This suggests that GMK exerts antioxidant activity, potentially preventing neuronal cell death caused by excitotoxicity. Values are expressed as the mean ± SD from three independent measurements. * *p* < 0.05 compared with the glutamate-only group, ** *p* < 0.01 compared with the glutamate-only group.

**Figure 5 cells-14-00977-f005:**
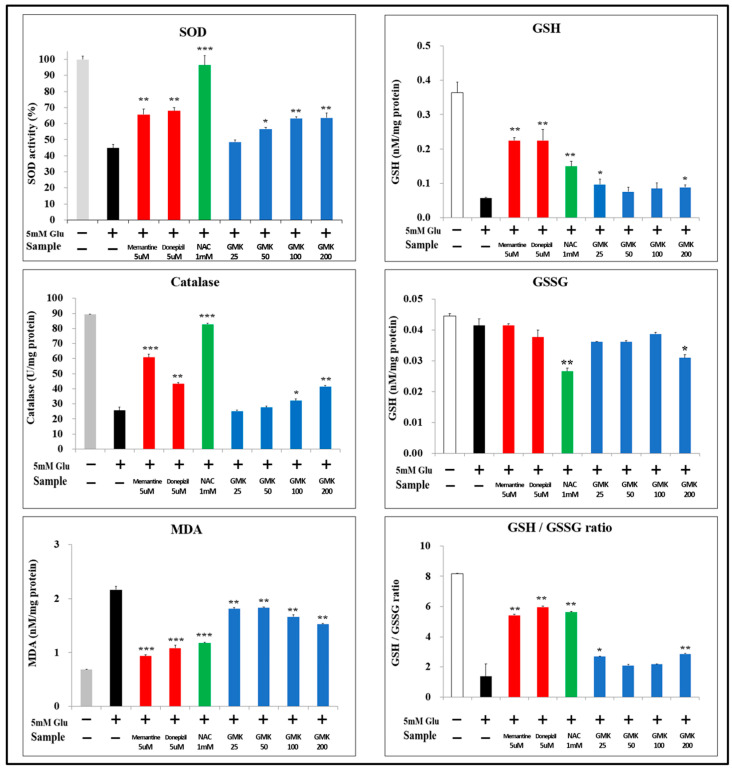
SOD, CAT, and GSH Levels. To determine whether the neuroprotective effects of GMK were attributable to its antioxidant properties, levels of SOD, CAT, GSH, and MDA were measured. GMK treatment resulted in a significant upregulation of SOD, CAT, and GSH levels, along with a significant downregulation of MDA levels. These findings indicate that GMK protects neuronal cells by reducing oxidative stress. Values are expressed as the mean ± SD. * *p* < 0.05 compared with the glutamate-only group, ** *p* < 0.01 compared with the glutamate-only group, *** *p* < 0.001 compared with the glutamate-only group.

**Figure 6 cells-14-00977-f006:**
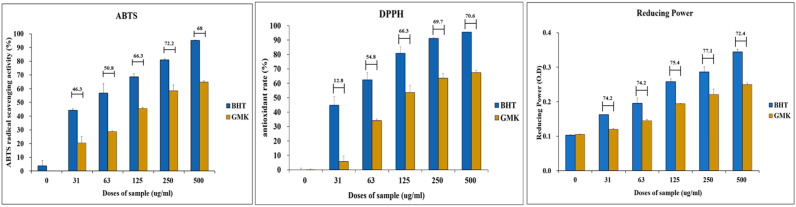
Antioxidant activity of the GMK hot water extract. Evaluation of the antioxidant activity of GMK using DPPH, ABTS, and reducing power assays revealed that GMK exhibited approximately 60% of the antioxidant activity of the positive control, butylated hydroxytoluene (BHT). The values shown above the bar graph represent the ratio of the reducing power of GMK to that of BHT at the same concentration.

**Figure 7 cells-14-00977-f007:**
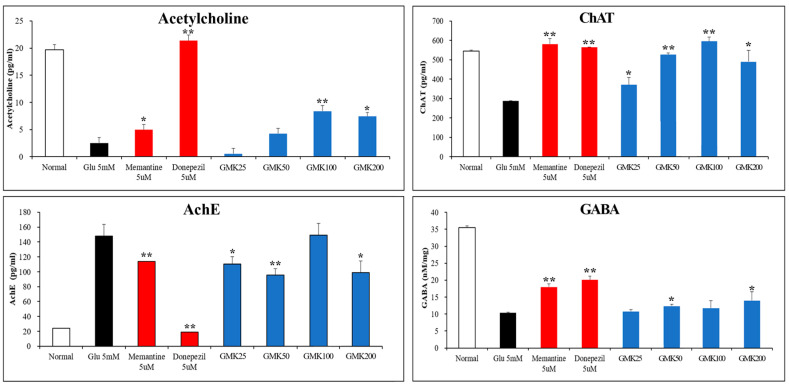
Enhancement activity of GMK on acetylcholine and GABA. GMK upregulated acetylcholine (ACh) and choline acetyltransferase (ChAT) levels while downregulating acetylcholinesterase (AChE) levels. Additionally, GMK increased GABA levels, suggesting that it may enhance inhibitory neurotransmission by improving the glutamate/GABA ratio and exerting inhibitory effects on excitatory signaling. Values are expressed as the mean ± SD. * *p* < 0.05 compared with the Glu 5 mM group, ** *p* < 0.01 compared with the Glu 5 mM group.

**Figure 8 cells-14-00977-f008:**
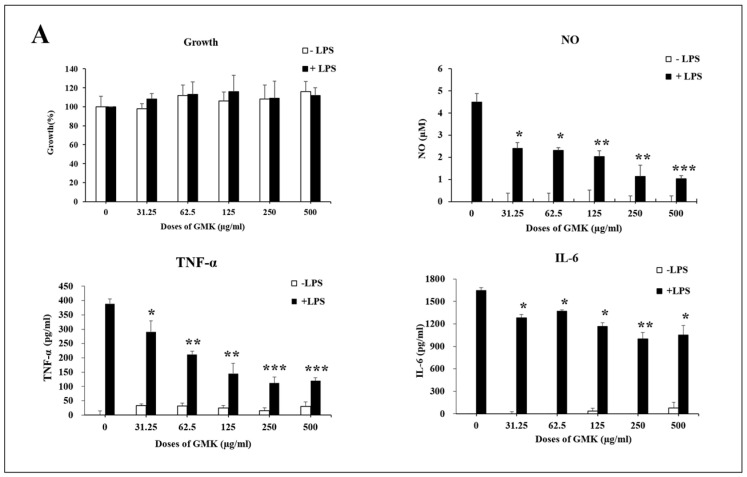
Anti-inflammatory (**A**) and anti-oxidative stress (**B**) activity of GMK in LPS-stimulated BV2 cells. GMK significantly attenuated the TLR4-mediated inflammatory response by inhibiting the phosphorylation of MAPKs, including p38 and JNK, as well as IκB, in LPS-stimulated microglial cells. Furthermore, GMK exhibited antiapoptotic properties by modulating the expression of BCL2 and BAX. It also demonstrated redox-regulating mechanisms through the downregulation of NOX1 and TXNIP. These effects of GMK resulted in the suppression of pro-inflammatory cytokines, such as TNF-α and IL-6, along with a reduction in nitric oxide production. These findings suggest that GMK has significant potential as a therapeutic agent for modulating inflammation and oxidative stress. Values are expressed as the mean ± SD. * *p* < 0.05 compared with the LPS group, ** *p* < 0.01 compared with the LPS group, *** *p* < 0.005 compared with the LPS group.

**Figure 9 cells-14-00977-f009:**
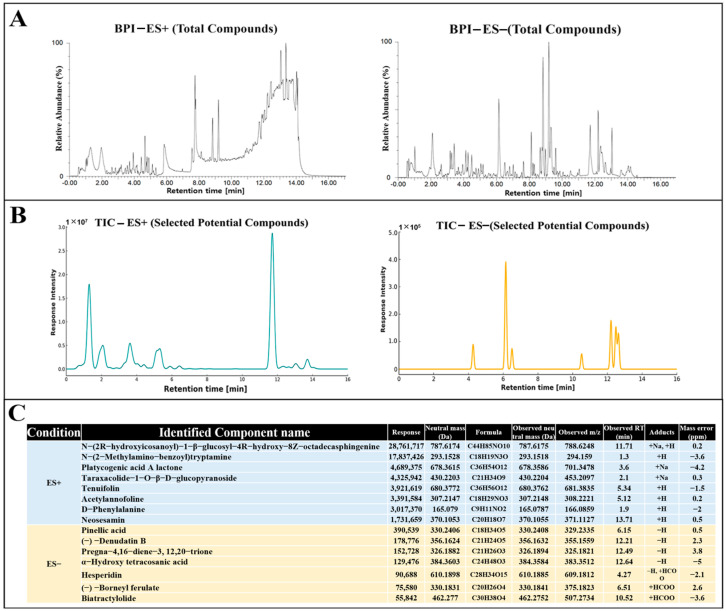
Tentative identification of compounds in GMK extracts by UHPLC-QTOF-MS/MS analysis. (**A**) Base peak intensity (BPI) chromatograms of GMK extract in positive (ES⁺) and negative (ES⁻) ionization modes. (**B**) Total ion chromatograms (TIC) of selected potential compounds in ES⁺ and ES⁻ modes. (**C**) List of identified candidate compounds based on UHPLC-QTOF-MS/MS analysis with corresponding mass data.

**Figure 10 cells-14-00977-f010:**
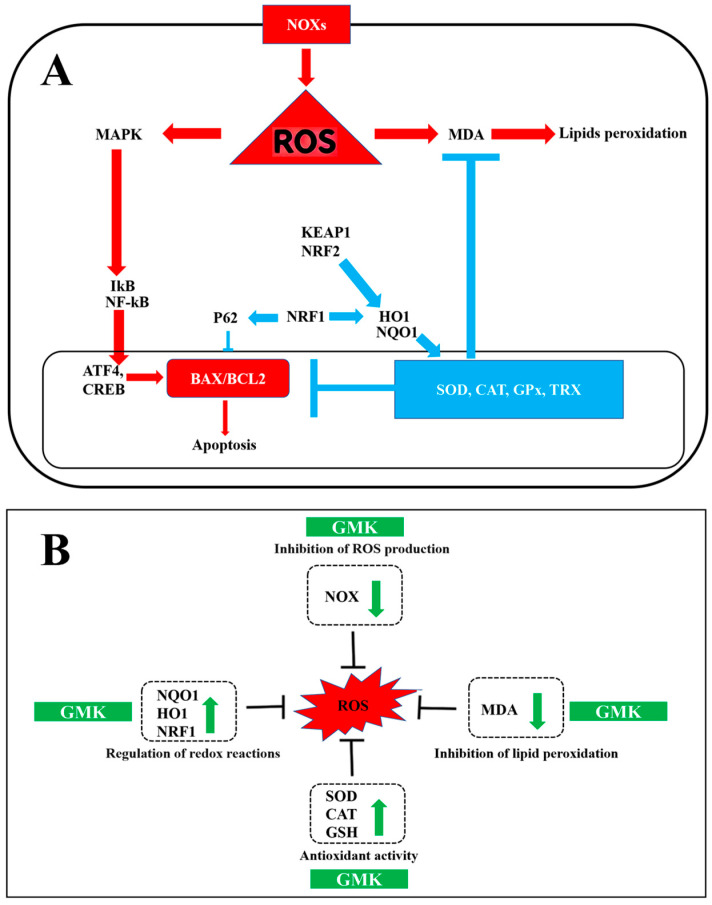
Schematic diagram summarizing excitotoxicity, inflammatory responses (**A**), and oxidation–reduction reactions (**B**), along with the effects of GMK.

## Data Availability

The original contributions presented in this study are included in the article. Further inquiries can be directed to the corresponding author.

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
