# Peer review of "Neuroprotective Effect of Mixed Mushroom Mycelia Extract on Neurotoxicity and Neuroinflammation via Regulation of ROS-Induced Oxidative Stress in PC12 and BV2 Cells"

_cells, 2025, doi:10.3390/cells14130977_

Round 1

Reviewer 1 Report

Comments and Suggestions for Authors

The authors reported the neuroprotective effects of GMK against neurotoxicity and neuroinflammation. They demonstrated that GMK can effectively counteract the deleterious effects of ROS through the modulation of various pathways. The authors concluded that GMK prevents excitotoxicity, inflammation, and redox reactions by removing excess ROS in an in vitro model. Overall, the concept of this manuscript is clear, and the experimental techniques used are adequate.

However, below is my comment on this manuscript that need to be addressed before publication:

  1. The concentration of GMK treatment is not consistent throughout the manuscript, and there is no clear explanation for the differences in treatment concentrations. For example:
  • Figures 2–4 show a GMK treatment range of 100–200 µg/ml
  • Figure 5: 25–200 µg/ml
  • Figure 6: 31–500 µg/ml
  • Figure 7: 50–100 µg/ml
  • Figure 8: 31.25–500 µg/ml

Please clarify how you initially selected the GMK concentrations and maintain the same concentrations throughout the manuscript. If changes in concentration are necessary, please provide a clear explanation for the increases or decreases. It is important to maintain consistent treatment concentrations across experiments or clearly explain the rationale for varying them. Consistency in treatment concentrations improves readability and facilitates comparison of results across different experiments.

Author Response

We sincerely thank the reviewer for this important suggestion. We have clarified the rationale for GMK concentration selection and variation across experimental platforms in the revised manuscript (Section 2.4: GMK Treatment Concentration Design). The overall concentration strategy is summarized below:

GMK Concentration Design

  Initial GMK concentration ranges were determined via MTT-based prescreening for cytotoxicity and subsequently optimized based on the analytical requirements of each assay (e.g., ELISA dynamic range, cytotoxicity threshold, or mechanistic signaling targets). In all experiments, the cytotoxicity threshold was defined as ≥80% cell viability, in accordance with the international standard ISO 10993-5 for in vitro cytotoxicity testing.

PC12 neurotoxicity model
  The 5 mM glutamate-induced MTT prescreen (Figure 1A) demonstrated that 100 and 200 µg/mL were the two most effective non-cytotoxic doses in both undifferentiated and NGF-differentiated PC12 cells. Accordingly, these concentrations were employed in all mechanistic PC12 experiments, including TUNEL staining, intracellular ROS quantification (FACS), and Western blotting.

PC12 antioxidant and neurotransmitter ELISAs
  Redox recovery involves multistep enzyme cascades that are highly sensitive to fluctuations in oxidative stress. ELISA-based detection of antioxidant markers such as SOD, CAT, GSH, GSSG, and MDA typically requires a broader concentration range to detect subtle dose-dependent changes [1]. To construct reliable four-parameter dose–response curves, we employed a four-point set (25, 50, 100, 200 µg/mL). Concentrations above 200 µg/mL were excluded because 400 µg/mL resulted in more than 15% reduction in cell viability.

  Although neurotransmitter regulation is redox-dependent via pathways such as NOX, NRF2, NQO, and xanthine oxidase, it also follows distinct neurochemical kinetics [2–4]. The initial concentration set used in Figure 7 (50 and 100 µg/mL) was selected to reflect functional modulation within a physiologically relevant range, rather than strict mechanistic equivalence [5]. However, in consideration of the reviewer’s valuable suggestion regarding consistency, we have now supplemented the dataset with additional measurements at 25 and 200 µg/mL, using thawed aliquots of previously collected supernatants stored at –80 °C.

BV2 inflammation model
  For NO, TNF-α, and IL-6 ELISAs, a broader two-fold dilution series (31.25–500 µg/mL) was applied, ensuring that all tested concentrations remained below the cytotoxicity threshold of 20% viability loss. As confirmed in Figure 8A, cell viability remained stable across this range. Western blotting for the BV2 model was then conducted at 100 and 200 µg/mL to match the PC12 mechanistic assays and enable direct cross-model comparisons.

Cell-free antioxidant assays
  Lastly, in cell-free antioxidant assays (DPPH, ABTS, and reducing power), GMK was tested at concentrations up to 500 µg/mL, as these assays are not constrained by cellular cytotoxicity. This broader range was selected to fully capture the radical scavenging and reducing capacity of GMK and to allow direct comparison with the positive control, BHT.

  1. Bibi Sadeer, N., Montesano, D., Albrizio, S., Zengin, G., & Mahomoodally, M. F. (2020). The Versatility of Antioxidant Assays in Food Science and Safety-Chemistry, Applications, Strengths, and Limitations. Antioxidants, 9(8), 709. https://doi.org/10.3390/antiox9080709
  2. Festerling, K., Can, K., Kügler, S., & Müller, M. (2020). Overshooting Subcellular Redox-Responses in Rett-Mouse Hippocampus during Neurotransmitter Stimulation. Cells9(12), 2539. https://doi.org/10.3390/cells9122539
  3. Ganguly, U., Kaur, U., Chakrabarti, S. S., Sharma, P., Agrawal, B. K., Saso, L., & Chakrabarti, S. (2021). Oxidative Stress, Neuroinflammation, and NADPH Oxidase: Implications in the Pathogenesis and Treatment of Alzheimer's Disease. Oxidative medicine and cellular longevity2021, 7086512. https://doi.org/10.1155/2021/7086512
  4. Kumar, A., Yegla, B., & Foster, T. C. (2018). Redox Signaling in Neurotransmission and Cognition During Aging. Antioxidants & redox signaling28(18), 1724–1745. https://doi.org/10.1089/ars.2017.7111
  5. Perluigi, M., Di Domenico, F., & Butterfield, D. A. (2024). Oxidative damage in neurodegeneration: roles in the pathogenesis and progression of Alzheimer disease. Physiological reviews104(1), 103–197. https://doi.org/10.1152/physrev.00030.2022

Reviewer 2 Report

Comments and Suggestions for Authors

In this well-performed study, the authors investigated the potential of a three-mushroom complex extract (GMK) to inhibit neuronal cell death induced by the activation of AMPA and NMDA receptors following glutamate treatment in NGF-differentiated PC12 neuronal cells. GMK significantly mitigated glutamate-induced excitotoxic neuronal apoptosis by reducing the elevated expression of BAX, a critical regulator of apoptosis, and restoring BCL2 levels. These neuroprotective effects were evidenced by the upregulation of SOD, CAT, and GSH levels, and the downregulation of MDA levels. The authors also revealed that GMK effectively scavenged ROS by downregulating NOX1, NOX2, and NOX4, while upregulating NRF1, P62, NRF2, HO1, and NQO1. Additionally, in the same model, GMK treatment increased acetylcholine, choline acetyltransferase, and GABA levels while reducing acetylcholinesterase activity. GMK also inhibited the activation of IκB and MAPK pathways, positively regulated the BCL2/BAX ratio, suppressed TXNIP activity, and upregulated NQO1 and NOX1. The authors concluded their study by stating that GMK could improve neuronal excitotoxicity and microglial inflammation through the positive modulation of the redox regulatory system, demonstrating its potential as a natural resource for pharmaceutical applications and functional health foods.

The paper is interesting and worth publishing. However, the authors should explain the rationale for the use of the 3 mushrooms used as an extract. Indeed, some doubts about their safety may be raised.

- Inonotus obliquus is traditionally grated into a fine powder and used to brew a beverage resembling coffee or tea, which tastes strongly of Chinese herbal tea. However, caution is warranted with chronic use due to the extremely high concentrations of oxalates (PMID 32419395).

- Ganoderma lucidum is inedible and rock-hard when dried, but is used to make a bitter-tasting tea, purported to have health effects by some cultures, although there is no reliable scientific evidence for such effects (ISBN 978-0-520-95360-4).

- Phellinus linteus is a mushroom known for its powerful antioxidant and anti-inflammatory properties, traditionally used in Asian medicine to support the immune system and fight infections. However, examining the available literature, it is noted that there is a lack of chronic toxicity studies and that there are still gaps in research regarding its mode of action and its pharmaceutical standardization.

As a minor point, the name of the species should always be written in Italics in any part of the text.

Author Response

We appreciate you for the reviewer's valuable comments on our manuscript. We carefully revised the manuscript based on the reviewer’s comments. The responses to the reviewer's comments are as follows, and the revised parts in the revised manuscript are marked in red.

Comment 1:
The paper is interesting and worth publishing. However, the authors should explain the rationale for the use of the 3 mushrooms used as an extract. Indeed, some doubts about their safety may be raised.

Inonotus obliquus is traditionally grated into a fine powder and used to brew a beverage resembling coffee or tea, which tastes strongly of Chinese herbal tea. However, caution is warranted with chronic use due to the extremely high concentrations of oxalates (PMID 32419395).

Ganoderma lucidum is inedible and rock-hard when dried, but is used to make a bitter-tasting tea, purported to have health effects by some cultures, although there is no reliable scientific evidence for such effects (ISBN 978-0-520-95360-4).

Phellinus linteus is a mushroom known for its powerful antioxidant and anti-inflammatory properties, traditionally used in Asian medicine to support the immune system and fight infections. However, examining the available literature, it is noted that there is a lack of chronic toxicity studies and that there are still gaps in research regarding its mode of action and its pharmaceutical standardization.

Response 1: Thank you for your comments regarding the preparation and safety of the research sample GMK. As described in the text, GMK was manufactured by simultaneously inoculating and cultivating three types of mushroom mycelia, including Inonotus obliquus, Ganoderma lucidum, Phellinus linteus, on a solid medium. The reason why these three types of mushroom mycelia were selected for GMK production is that they have a very high variety of physiological activities and have been used for a long time in traditional medicine in Asian countries such as Korea, China, and Japan. We expected that if these three types of mycelia were cultured simultaneously, various active ingredients would be produced through antagonistic interactions during the culture process, thereby obtaining a mushroom mycelia mixture with high physiological activity. In fact, in anti-inflammatory and anti-diabetic effects, GMK was confirmed to have significantly higher activity compared to each mushroom mycelia alone or a simple mixture of these mycelia. For this reason, we have been conducting studies on the inhibitory activity against inflammatory diseases and metabolic syndrome using GMK, a mixed mushroom mycelia extract of three types. Regarding the safety of GMK, its safety has already been confirmed through toxicity tests such as single-dose toxicity, genotoxicity, and 13-week continuous administration by an authorized analysis agency approved by the Korean government.

Comment 2: As a minor point, the name of the species should always be written in Italics in any part of the text.

Response 2: As pointed out by the reviewer, the names of the species of mushrooms have been italicized.

Round 2

Reviewer 1 Report

Comments and Suggestions for Authors

Thank you for reply my comments. The author has satisfactorily responded to all of my feedback, I am happy to recommend this paper for publication.

Reviewer 2 Report

Comments and Suggestions for Authors

Nice paper